Public domain. CC0 1.0.

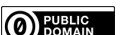


# Evaluating mass flow meter measurements from chambers for greenhouse gas emission from orphan wells and other point sources

Karl B. Haase[1] and Nicholas J. Gianoutsos[2]

[1:] khaase@usgs.gov, U.S. Geological Survey Geology, Energy & Minerals Science Center, 12201 Sunrise Valley Drive, MS
432, Reston, VA 20192
[2:] ngianoutsos@usgs.gov, U.S. Geological Survey Energy Resources Program, Central Energy Resources Science Center, Box 25046, MS 939, Denver, CO 80255

*Correspondence to*: Karl B. Haase (khaase@usgs.gov)

**Abstract.** This study evaluates the performance of a rigid gas flux chamber equipped with a mass flow meter (MFM) for
measuring gas emissions from leaking orphan wells and similar pressure-driven gas point sources. We conducted a series of laboratory and field experiments to evaluate the sensitivity, stability, and dynamic range of an MFM chamber system and found an optimal method for sealing the chamber to the ground to isolate the emission source. From these results, we estimate the effects of different soil gas permeabilities on measurements and identify the uncertainty of environmental processes that can impact measurements. Simulations of an MFM chamber are compared to those of a dynamic flux chamber to contrast the
data derived with both methodologies and illustrate the potential for measuring high variability leaks with the MFM chamber. Using a low flow resistance MFM and a chamber well-sealed to the ground, it is possible to measure leaks down to $1.08 \times 10^{-3}$ cubic meters per hour (m$^3$/h) (0 °C/1 atm; STP), corresponding to 0.77 grams per hour (g/h) methane or 2.11 g/h carbon dioxide, with a mean uncertainty of 0.89 % relative standard deviation. Environmental processes such as heated gas inside the chamber from solar gain, wind blowing across the chamber vent, and changing humidity in the chamber, can cause variation
in MFM measurements. Over 11 days of continuous monitoring under varying weather conditions, the standard deviation of the environmentally sourced signals was found to be 7.40 10$^{-3}$ m$^3$/h (equivalent to or 5.27 g/h methane or 14.45 g/h carbon dioxide). Strategies to obtain the highest quality data from MFM chambers include burying the edges of the chamber below the surface sufficiently deep to seal the chamber edges against gas flow and soaking the dirt with water to lower the chances of escaping gases, while monitoring the gas flow and adjusting the chamber seal to achieve a maximum flow rate.

## 1 Introduction

Over the past few years, interest in methane emissions from orphan and abandoned oil and gas wells has increased, presenting an opportunity to examine the mechanisms of leaking wells and, ultimately, reduce greenhouse gas emissions from sources that do not provide any economic, social, or ecosystem benefit (Alboiu and Walker, 2019; Kang et al., 2023). Over the past decade, studies quantifying methane emissions from orphan and abandoned wells have been conducted by many researchers
(Kang et al., 2014; Kang et al., 2016; Townsend-Small et al., 2016; Pekney et al., 2018; Riddick et al., 2019; Lebel et al., 2020; Williams et al., 2019; Saint-Vincent et al., 2020b; Townsend-Small and Hoschouer, 2021; Nivitanont et al., 2023). The passage of the 2021 Infrastructure Investment and Jobs Act (IIJA) provided 4.7 billion dollars to states, federal land management agencies, tribes, and other government entities to plug orphan wells (U.S. Public Law 117-58, 117th Congress). A requirement to receive federal funding to plug orphan wells was to screen them for methane leaks, and if found, conduct methane emissions
measurement before plugging, which is summarized for Congress (Department of the Interior Orphaned Wells Program Office,

Public domain. CC0 1.0.

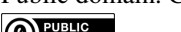


2023). For wells plugged with non-IIJA related funding, there is interest by private, as well as state organizations, in using the emissions reduction from plugging orphan wells to create carbon credits (CarbonPath, 2024; BCarbon, 2024; American Carbon Registry, 2023; Zer0six, 2024). This has spurred interest in accurately measuring methane emissions from orphan wells to meet the measurement requirements from federal plugging programs, evaluate emissions for calculating carbon credits, and to

better understand overall greenhouse emissions (Williams et al., 2021; Kang et al., 2023).

Measurement of emissions from orphan wells is challenging for several reasons. The emissions distribution from orphan wells is strongly skewed, with the bulk of wells having extremely low emissions of methane (less than one gram per hour (g/h)) while there are relatively few wells that emit methane in the range of hundreds of grams per hour or more (Williams et al., 2021) and a handful have methane emissions rates over 1 kilogram per hour (Bowman et al., 2023; Riddick et al., 2024).

The physical geometry of a well and the specific locations of leaks can pose a challenge for measuring emissions. Each well must be screened before emissions are measured and a strategy for measuring all leaks must be developed. Orphan well emissions are difficult to distinguish using areal integration and remote sensing techniques (i.e. eddy covariance flux and plume-based approaches) because they are often of a very low magnitude relative to other greenhouse gas sources on the landscape, and they often necessitate on-site strategies to definitively isolate their emissions. Further complicating the matter,

emissions from orphan wells are known to vary over time, and there is need for time series emissions data (spanning days or longer) with a measurement resolution of minutes to hours (Riddick et al., 2020). Time series measurements can be complex due to the power requirements of most analytical systems (Riddick et al., 2022). Due to the typically small emissions rate of leaking orphan wells and the current technological challenges, broader range emissions monitoring approaches and measurement strategies could help to address the problems associated with orphan well emission monitoring.


### 1.1 Aims of this investigation

The purpose of this study is to demonstrate the performance of a rigid flux chamber with a thermal mass flow meter (MFM) in response to controlled gas flows and to illustrate how deployment conditions of the measurement system, gas composition of the emissions, and quality of the chamber seal against the ground affect measurement accuracy and variability. In this study,

a series of controlled laboratory and field tests were conducted where compressed air was supplied to an MFM chamber at known flow rates (Fig. 1 a) to emulate the mechanics of the system when placed over emission sources like orphan wells (Fig. 1 b) and other point sources, such as mine and landfill vents and cave entrances. This study also presents measurements of meteorological and environmental processes that can create flow within the MFM chamber and estimates the impact of gas loss through permeable soils. The goal of this work is to optimize MFM chamber measurements, identify common mistakes

that can cause underrepresentation of emissions, and ultimately identify opportunities for future improvements in implementation.

Public domain. CC0 1.0.



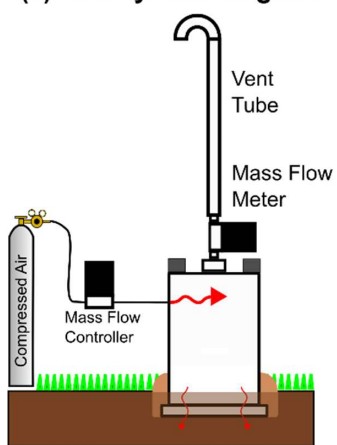

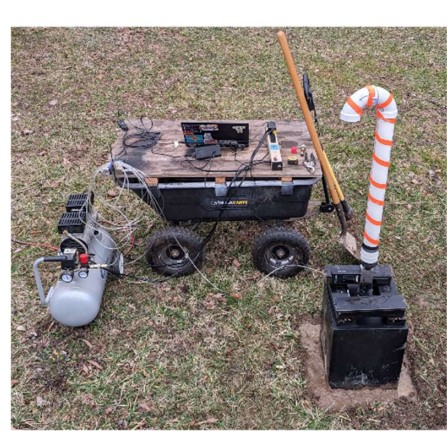

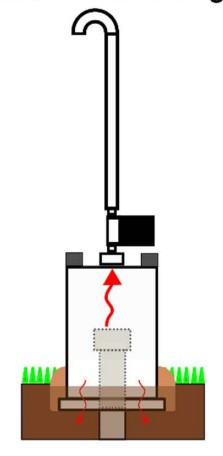

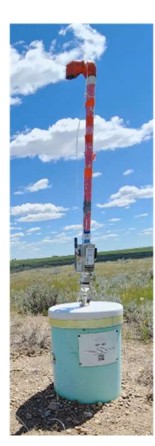

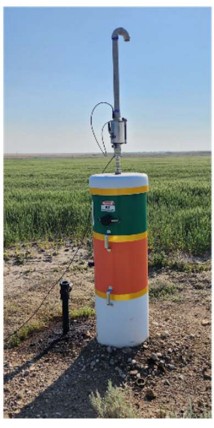

**Figure 1: Overview of the test system used to emulate emissions in comparison to the configuration used to measure emissions in the field. (a) A conceptual diagram alongside (b) a mass flow meter (MFM) test system setup in the field. (c) a diagram of the deployment of MFM measurement systems quantifying orphan wells emissions and (d) photos showing measurement systems with different size chambers and vent pipe configurations. (Photo sources: Karl Haase (c) and Nicholas Gianoutsos (d), USGS).**

## 2 Overview of measurement approaches

Measuring orphan well emissions is challenging compared to other emission sources because the wells are often in a state of disrepair, which can make it difficult to directly connect a measurement system to the leak source. There are a range of methods in the literature for measuring gas emissions from weak point sources like orphan and abandoned wells, landfill vents,



abandoned coal mines, and sewer caps (Mønster et al., 2019; Joo et al., 2024; U.S. Environmental Protection Agency, 2004). Riddick et al. (Riddick et al., 2024) summarized many methods used for emission measurements from orphan and abandoned oil and gas wells, which typically include static chambers (Kang et al., 2014; Kang et al., 2016; Townsend-Small et al., 2016;

Lebel et al. 2020; Townsend-Small and Hoschouer, 2021; El Hachem and Kang, 2022), dynamic chambers (Townsend-Small et al., 2016; Pekney et al., 2018; Riddick et al., 2019; Riddick et al., 2020; Townsend-Small and Hoschouer, 2021), high flow samplers (Townsend-Small et al., 2016; Pekney et al., 2018; Saint-Vincent et al., 2020a; Townsend-Small and Hoschouer, 2021), and plume based approaches (Atherton et al., 2017; Lebel et al.; Vogt et al., 2022; Riddick et al., 2024). A key theme among these approaches is that they are based on concentration measurements and that any required gas flows are generated

by pumps and blowers. This requirement increases the physical and power footprints of the systems, adds technical complexity, and serves as a barrier for use in time series emissions measurements (Fleming et al., 2021).

        A contrasting approach demonstrated by DiGiulio et al. (2023) and Follansbee et al. (2024), and accepted by the American Carbon Registry (American Carbon Registry, 2023) and the U.S. Department of Energy (DOE) (Pekney et al., 2024), is time series measurements of the total gas flow from the leak source that is then corrected for the concentration of the

greenhouse gas in question. For wells (such as orphan wells with multiple leaks or missing surface casing) and other point sources that cannot be directly connected to an MFM, a chamber is placed over the orphan well and a seal is formed with the ground to prevent emissions from escaping the chamber (Fig. 1 b). The chamber is vented to the atmosphere through a low-pressure drop, high dynamic range MFM and pipe stack to prevent water and particle ingress and back diffusion of gas to the MFM. The mass flow of gases leaving the chamber is either read directly from the meter by the user or logged over time by

an external data capture system. This process resembles measurements made from gas flows through landfill covers (Maciel and Jucá, 2000; Zhan et al., 2016) and soil gas permeability (Allaire et al., 2008) that use both gas concentration and chamber flow to calculate gas flux. MFMs are used to quantify emissions from surface casing vent (SCV) ports, well casings, and leaks from valves and ports on the wellhead where the flow meter can be directly attached to measure emissions (Alberta Energy Regulator, 2022). Attaching an MFM to a rigid chamber expands the range of scenarios where an MFM can be used because

the entire wellhead is enclosed, encompassing all the leak pathways and acting as an extension of the wellbore above the ground. This approach is different from other chamber-based approaches because the chamber is constructed and operated with the intent of maintaining a pressure differential between the chamber and the atmosphere, while other approaches seek to minimize the pressure differential (Maier et al., 2022; Thalasso et al., 2023; Riddick et al., 2023; Williams et al., 2023). There are several conceptual benefits of this approach, including that the flux measurement can be made at a wide range of flow

rates, limited by the dynamic range of an MFM, without any moving parts, and the response time of the measurements is not limited by the mixing time of the chamber. MFMs have several characteristics that make them appealing for long-term measurements in the field: they have stable response factors, with calibration frequencies of a year or more commercially available, and some are available with ATEX (ATmosphères EXplosibles) and IECEx (International Electrotechnical Commission for Explosive Atmospheres) certifications for operation where explosive mixtures may be present. Drawbacks to

using MFMs include different responses to gas mixtures that are a function of both the thermodynamic properties of the gas



being measured and the MFM internal geometry. For maximum accuracy, the gas mass flow measurements can be corrected for composition with external analysis to correct the mass flow measurements for air dilution (DiGiulio et al. (2023).

## 3 Conceptual Model of the Measurement System

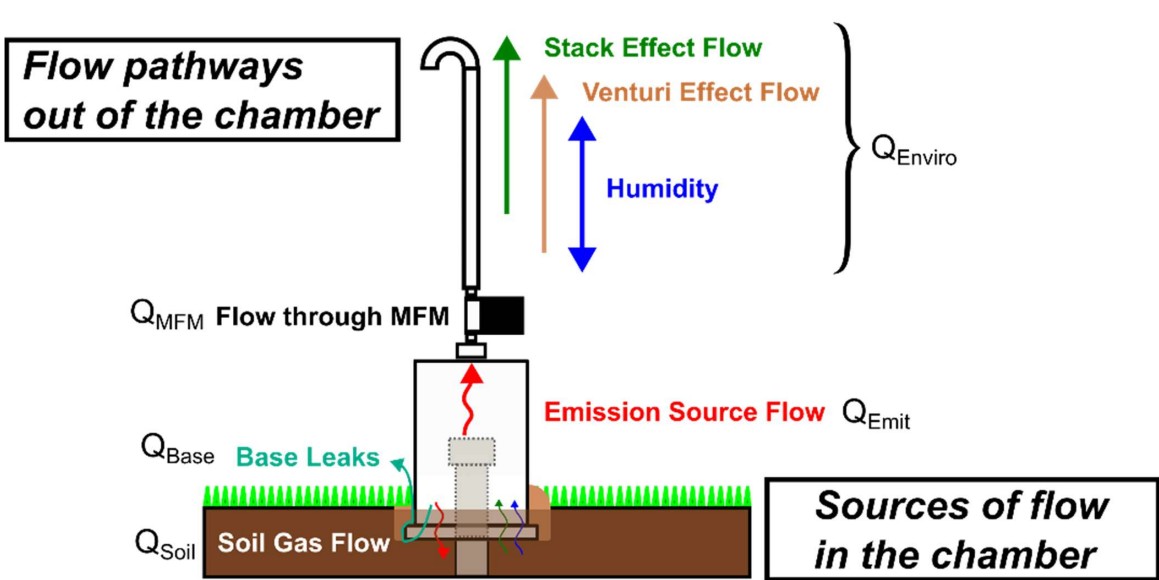

**Figure 2: Schematic of flow sources and flow paths in a mass flow chamber flux measurement system. Contributions to gas flow through the chamber include the gas emission source (red) and the sources of environmental flow examined in this study, including the stack effect (green from solar heating of the chamber), the Venturi effect flow from wind (tan), and humidity changes (blue) from water evaporating and condensing in the chamber and include changes in water vapor pressure resulting in different mass flow meter (MFM) responses. Additionally, there are gas flow pathways out of the chamber, including through the MFM, through**

**the soil (small arrows for movement of gases through the soil), and through leaks at the soil-chamber interface at the base (turquoise).**

Figure 2 illustrates the different sources of gas flow in the chamber and the flow pathways out of the chamber investigated in this study. The gas flows out of the chamber are composed as the sum of: $Q_{MFM}$, the amount of gas flow rate through the MFM, $Q_{Base}$, the gas flow rate through leaks at the interface between the chamber and the ground, and $Q_{Soil}$, the gas flow rate through

the soil or surface beneath the chamber. These flows are in response to gas released into the chamber from the leak being measured, $Q_{Emit}$, and changes in flow through the chamber caused by environmental processes ($Q_{Enviro}$), yielding:

$$Q_{MFM} + Q_{Base} + Q_{Soil} = Q_{Emit} + Q_{Enviro} \qquad (1)$$

In the ideal case, the mass flow of gases into the chamber is dominated by emissions from the well or point source over which the chamber is placed ($Q_{Emit}$). Practically, gas flows can be modulated by environmental processes ($Q_{Enviro}$) that create pressure

Public domain. CC0 1.0.

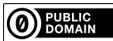



deviations between the chamber and the outside environment, as explored below. Some examples include evaporation of water

and evolution of other gases in the soil underneath the chamber, stack effect from temperature differences between the

atmosphere and the air in the chamber, the Venturi effect of wind blowing across the MFM vent, barometric pumping between

the ground and atmosphere across the chamber, and gas flow through the soil caused by soil tension and/or changes in the

water table from movement of water through the unsaturated zone during precipitation (Maier et al., 2019, Yang, et al., 2021,

Li, et al., 2022).

        Here, we evaluate the contributions of $Q_{Base}$ and $Q_{Soil}$ on the accuracy of the measurements of the MFM chamber.

$Q_{Base}$ is dependent on the procedures and environmental materials used to seal the chamber to the ground. The contact area

between the rigid chamber wall and the ground can create preferential flow paths that contribute to the total gas conductance

of the chamber by allowing gas to flow out of the chamber system, so it is critical that $Q_{Base}$ be made negligible though an

effective ground seal. $Q_{Soil}$ is an important parameter because it is inherent in the measurement site and is only partially

controlled by chamber design and measurement protocol. $Q_{Soil}$ can be described by Darcy's law, defined by the following: (1)

the chamber footprint area ($A$), (2) the length ($L$) of the gas flow path from the soil surface in the chamber to the soil surface

outside the chamber, (3) the gas permeability of the soil to the gas in the chamber ($k_g$), (4) the dynamic viscosity of the gas in

the chamber ($u_g$), and (5) the pressure difference between the chamber and the atmosphere ($\Delta P$), (Bloomfield and Williams,

1995; Yang et al., 2019; Garg et al., 2021) yielding equation 2:

$$Q_{Soil} = \frac{A \times k_g \times \Delta P}{(u_g \times L)} \quad (2)$$

Thus, in field scenarios, there are situations that increase $Q_{Soil}$, such as shallow chamber placement that reduces L or placement

of a chamber in permeable soils with high $k_g$. It is also important to note that $u_g$ is not known *a priori*, and varies depending

upon the chemical composition of gases being released to the chamber, as dynamic viscosity changes with gas composition,

for example, with methane being about 40 % and $CO_2$ being 20 % less viscous than air, respectively (Friend et al., 1989;

Laesecke and Muzny, 2017). Finally, the signal from the MFM may not report the true emission rate of the gas in question, as

the response of the MFM changes with gas composition and concentration corrections need to be applied in practice to

determine both the true value of $Q_{MFM}$ and the composite flux of gases from the source.

Public domain. CC0 1.0.



## 4 Methods:

**4.1 Test Chamber Design, Operation, and Field Tests**

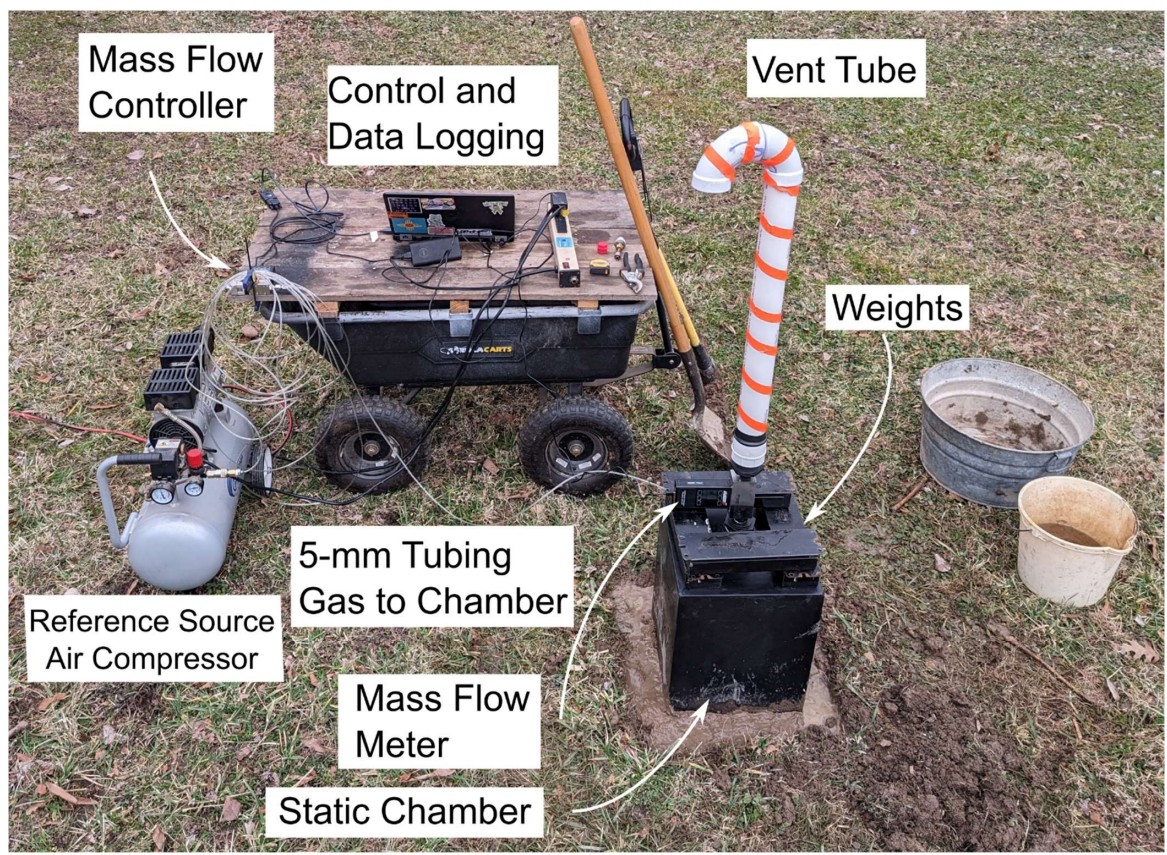

**Figure 3: A photograph of the mass flow meter (MFM) chamber system under test with the major system components labelled. The reference air source is provided by an air compressor and mass flow controller. (Photo source: Karl Haase (USGS) )**


An MFM chamber testing system was constructed as shown in Fig. 3. The dimensions of the rigid chamber were 30 cm × 30 cm × 45 cm, and all flow measurements used a 0-100 standard liters per minute (slpm) low pressure drop MFM. The MFM vent length was 1 m. For some tests, a high density polyethylene (HDPE) plastic ground skirt was attached to assess the feasibility of a passive seal at the ground surface that did not require digging for placement (Parkin et al., 2005; Thalasso et



al., 2023). The reference gas was ambient air delivered via an oil-free air compressor to a 0-5 slpm (0-0.3 m$^3$/h) mass flow controller (MFC), which permits large flow rates to be tested for extended periods of time in safety from combustion or asphyxiation and without the associated costs of compressed pure gases. The MFC dynamic range allows mass flow rates of air equivalent to be added to the chamber that are equivalent to 0.02 to 214.5 g/h of methane and 0.06 to 589.2 g/h of $CO_2$. As the method focuses on advective flows, selective or diffusive transport into the polymers of the chamber are assumed to be

negligible relative to the flow through the MFM. The control and data recording loop for the MFM and MFC operated at rates of 1-10 s$^{-1}$. Additional sensors and data sources for humidity, temperature, wind, solar flux, and pressure were mounted in the vent stack around the chamber during evaluation of environmental factors. Details of construction and operation of the chamber are given in the supplementary information (SI) and the data collected in the study are available as a U.S. Geological Survey (USGS) Data Release (Haase and Gianoutsos, 2025).

Chamber field testing was performed on soils near the USGS headquarters in Reston, Virginia, USA. Soil plots were characteristic of domestic sod and mowed meadows in the area, with a mix of local and non-native sedges. The soil type is Chantilly-Dulles complex, which is generally high clay and low water permeability (Northern Virginia Soil and Water Conservation District, 2013). Studies were carried out in the field between December 2023 and February 2024 in varying weather conditions.

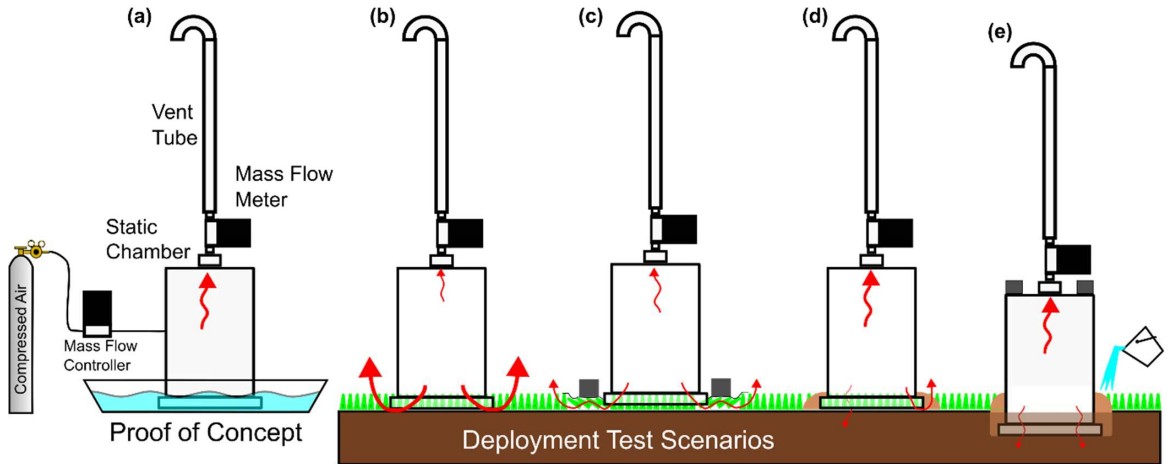


**Figure 4: Diagram of flow test scenarios. (a) chamber in water bath to validate performance with perfect base seal and no leaks. (b) chamber on ground surface (c) chamber sealed to ground with weighted polyethylene skirt (d) chamber on ground surface sealed with dirt, (e) chamber buried at various depth and sealed with dirt and added water to improve soil compaction and ground seal formation. Grey squares are weights to hold the chamber down. Arrow widths denoted the relative amount of gas flows through**
**different paths.**

     The MFM was first tested in a pool of water to provide a reference point for a tight seal and to ensure that the chamber and flow system was otherwise leak free. Control tests were then conducted with a range of base seal conditions to ascertain

Public domain. CC0 1.0.

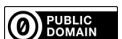



how best to seal the chamber to the ground (Fig. 4). These included simply setting the chamber on the ground (Fig. 4 (b)), attaching a HDPE plastic skirt to the edges of the chamber and weighting it to seal against the ground surface (Fig. 4 c), pilling

dirt on the edges of the chamber to form a seal (Fig. 4 d), and digging trenches of various depths and packing them with dirt and wetting the dirt in the trenches to loosen soil and reduce pore volumes, and adding weight to the chamber to prevent it from shifting (Fig. 4 e). After each seal was established, a reference flow test over the entire MFC range was conducted. Before each test, the MFM on the chamber was zeroed with the outlet plugged. All reference flow tests were conducted over a range of controlled flows between (0-0.3 m$^3$/h). During the reference flow tests, the MFM was measured with no reference flow,

then the reference flows were initiated at the highest flow rate (0.3 m$^3$/h), and then sequentially decreased through 9 steps to $6.0 \times 10^{-5}$ m$^3$/h before a step at zero flow, with a flow duration of three minutes per step. The pressure response of the chamber is not instantaneous, and one minute of data was collected after the signal of the MFM had stabilized and was used for tabulating the MFM output. All flow rates in this study are reported at reference conditions of 25 ℃, and 1 atm.

After a sealing protocol was developed, three blind tests were conducted where the chamber was sequentially

deployed in new locations a few meters apart and the reference flow sequence was executed to evaluate its effectiveness and reproducibility. Environmental testing to evaluate the contributing environmental processes on the MFM chamber measurement took place in a range of weather conditions (hot/cold, cloudy/sunny, windy/calm, and precipitation). These tests were conducted both with a plastic skirt and the chamber flat on the ground, and with the edges of the chamber sealed to the ground with packed dirt. Data were collected from the control computer daily during this period and, afterwards, the vent port

of the chamber was plugged and the MFM was re-zeroed. This is important to note because it likely had the effect of muting the contribution of changes in relative humidity on the MFM data stream but was important to remove baseline drift in the MFM (which could also be a response of the MFM electronics to temperature, battery voltage, dust on the MFM sensing elements, or other sources). Data collected during environmental process testing were evaluated for quality, and data from periods when the sensors were not operating correctly (typically power failure to a component or the system, downtime to

retrieve data, unresolved gas reference flows, and condensation in the MFM) were removed from the data stream.

### 4.2 Laboratory Testing of Mass Flow Meter and Chamber Assembly

To validate linearity of the reference flow MFC and the chamber MFM devices, both were connected in series (without the chamber in the middle) and the reference gas flow test routine was run over the full MFC range (i.e. to show that $Q_{emit} = -Q_{MFM}$ without a chamber or soil losses). To assess the precision and stability of the MFM chamber system response in a controlled

environment, free from wind, temperature changes, and direct sun, the sealed chamber was tested with over 42 measurement cycles spanning a 15-hour period. For this test, the bottom of the chamber was sealed with 152-μm-thick high-density polyethylene (HDPE) sheet taped to the chamber edges.

Public domain. CC0 1.0.



### 4.3 Chamber Simulations: Soil Permeability, Gas Composition, and Response of MFM compared to Dynamic Chambers

During field visits where the MFM was mounted on chambers placed over orphan wells (Haase, et al. 2025), we observed that methane was detectable around the base of the chambers over wells with methane leak rates of over 100 g/h range, showing that gas can migrate through the soil even where the chamber has been placed with the best possible ground seal. The effect of gas flow through the soil under the chamber on measurement accuracy, soil gas flows were estimated for a range of soil permeabilities using equation 2. Soil gas flows were calculated for a range of mass flow meter pressure drops collected from

various vendor's datasheets.

A second set of calculations were used to evaluate the effect of the concentration dependence of the MFM on emissions rate measurement accuracy. For this evaluation, both an MFM chamber and an identically sized dynamic flux chamber were simulated. The simulated chamber calculations assumed a volume of 250 L, with a pumping speed of 5 liters per minute (L/min) in dynamic chamber mode for 24 h. Stable methane leaks of $1.40 \times 10^{-3}$ m$^3$/h (1 g/h CH$_4$), $6.00 \times 10^{-2}$ m$^3$/h (42

g/h CH$_4$) and pulses every 30 minutes (1 minute at $6.00 \times 10^{-2}$ m$^3$/h and 1 minute at $3.00 \times 10^{-2}$ m$^3$/h to emulate burst and pressure decay) were simulated. The initial conditions for the chamber were set to 1 part per million of methane and the balance air. The simulations calculated the concentration change of gas in the chamber at each time step, then they calculated what instantaneous leak rate each measurement system would report. For the MFM chamber, the simulated emission rate measurement assumed instantaneous response to flow changes, and the simulated MFM response was referenced to pure

methane, with air to a methane response factor of 1.367 (MKS Instruments, 2024). For the dynamic chamber, the simulated emission rate measurement was calculated using the following equation, following (Riddick et al., 2023, Wiliams 2023):

$$Q_{leak}=(X_{chamber}-X_{background})*R_{pump} \qquad (3)$$

where $X_{chamber}$ is the steady state volume mixing ratio of the gas of interest in the chamber, $X_{background}$ is the background mixing ratio of the gas in air, and $R_{pump}$ is the mass flow rate of gas being removed from the dynamic chamber by the pump.

## 5 Results: Testing Deployment Protocols in the Laboratory and the Field

### 5.1 Sealing the Chamber Against the Ground

Many different approaches to sealing the chamber against the ground were tested (Fig. 4 and the methods section). A reference point was created by testing the chamber when it was placed in the pool of water (Fig. 4 a). This test showed good linearity but a slow response to changes in flow as increasing air pressure pushed water out of the chamber. Additionally, the wave

action of water in the pool generated noise in the response as the water moved in and out of the chamber. Gas flow could not be measured from approaches that did not have a direct seal to the ground or thin soil barriers as these allowed small, uncorrectable leaks around the chamber edges. Tests with the chamber resting directly on the ground (Fig. 4 b), with a HDPE plastic skirt weighted with chain and bricks (Fig. 4 c), and with soil packed against the edge (Fig. 4 d) could not be configured to reliably generate a useful signal for reference flow under any condition, as the gaps formed between the edge of the chamber,

Public domain. CC0 1.0.

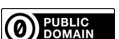



the sheet, surficial debris, and the packed dirt allowed a major portion of the reference gas flow to leave the chamber. Burying the edges of the chamber at a shallow depth of $\approx$ 3 cm and packing soil around the edge of the chamber would occasionally generate a seal that would allow full response over the test, but, over time, small seal defects could appear as the soil settled (Fig. 5), making it difficult to complete a full reference flow test. Wetting the edges increased the quality of the seal, as did increasing the depth of the trench. Ultimately, it was determined that a chamber placement depth of 18 cm (7 inches) or more,

removing fibrous and porous debris from the soil before packing (roots, sticks, rocks, and pebbles, ect.) and wetting the soil packed at the edges, provided a reliable seal that could be blind tested against reference flows and could achieve reliable results. Figure 6 provides example results from flow tests with the MFM chamber placed in a pool of water in the field (to demonstrate a leak-free seal), tests from a leaking chamber in shallow soil, and a test from a tightly sealed chamber deep in the soil.

260        The results of different controlled flow experiments where a good seal was achieved are shown in Table 1, and the comparison of the reference direct MFC-MFM connection, water bath tests, and soil placement tests linearity and accuracy tests are shown in Fig. 7. Three repeated placement tests at 18 cm depth showed that the chamber could be placed reliably, and the response of the MFM relative to the MFC reference flow was stable and linear, ranging from 1.0042 to 1.0125, with detection limits (3 $\sigma$) of 8.18 $\times10^{-3}$ to 1.385 $\times10^{-2}$ m$^3$/h, corresponding to a pure methane leak rate of 0.58 to 0.99 g/h. The

average uncertainty (average of the relative standard deviations across the flow range) was in the range of 1% for all tests, though the uncertainty was higher for the lower flow rates, increasing beyond 10% at below 1 $\times10^{-2}$ m$^3$/h, with variability due to environmentally driven flow (as discussed below) and likely due to variation in the quality of the seal between the placements. It is important to note the detection limit of gas flows and the detection limit for specific gases emission rates are separate: If a gas source is leaking a mixture of a gases, then the detection limit is reduced by the fraction of the specific gas

in the flow, for example if the flow limit of detection is ~1 $\times10^{-2}$ m$^3$/h, but the concentration of methane is 1%, then the methane limit of detection would be 1 $\times10^{-4}$ m$^3$/h, or 7.4 $\times10^{-3}$ g/h.

Public domain. CC0 1.0.



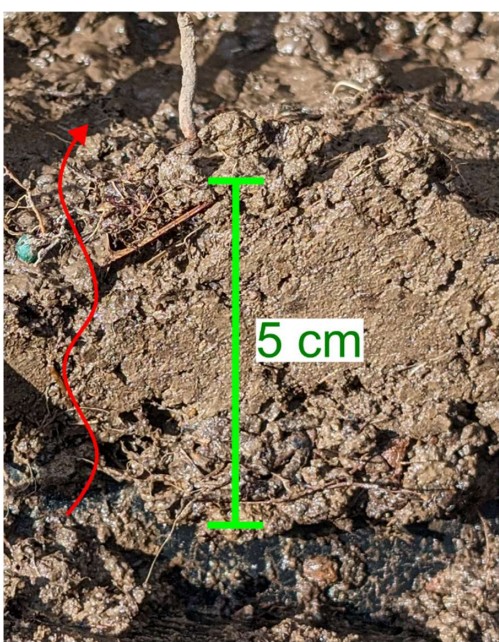

**Figure 5: A photograph of a cross section of local soil that was packed against the side of the chamber after a deployment test that did not completely seal. A potential gas leak pathway (red lines) following roots in the soil. Rocks, roots, sticks, and other materials can be present in the soil that create a preferential flow path for gas to flow around the edges of the MFM chamber. (Photo source: Karl Haase (USGS) )**




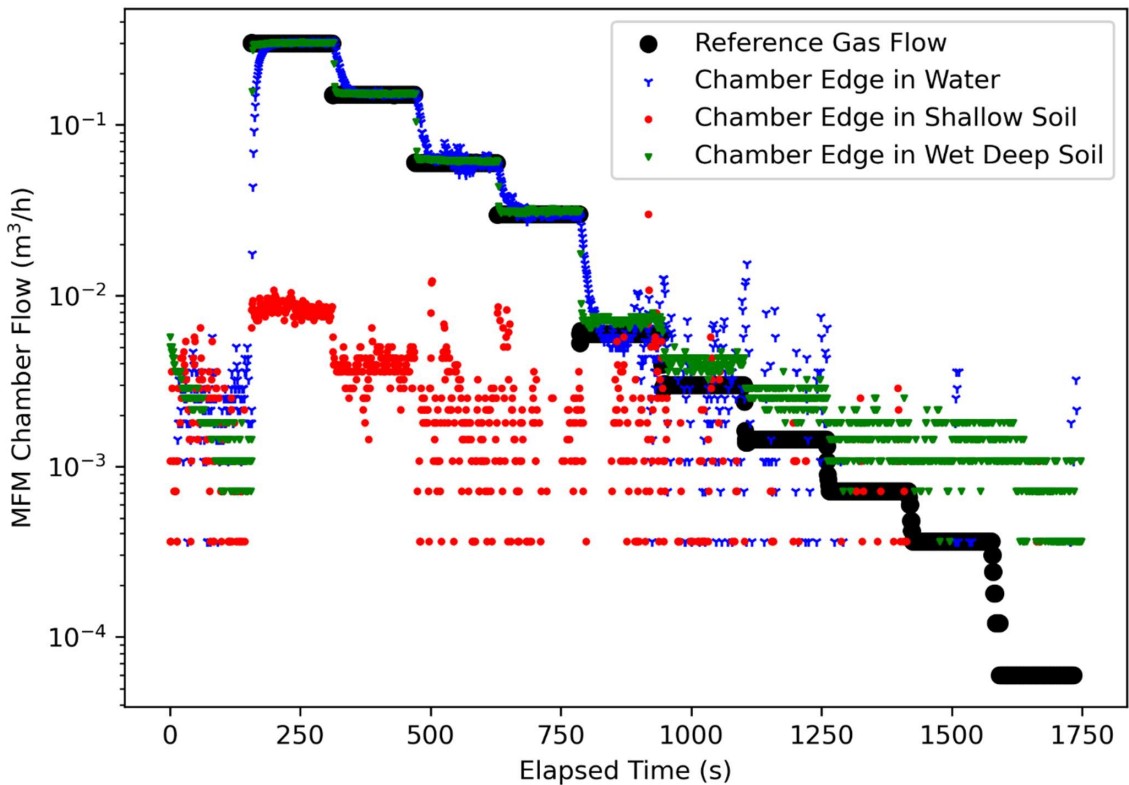

**Figure 6: Mass flow meter (MFM) signal collected with the chamber under different test conditions with different qualities of ground**
**seals. All tests were conducted outdoors and noise from wind is present. The reference air flow into the chamber (black circles)**
**represents the best-case baseline. MFM signal with chamber placed in water (blue three-point marker), shows noise from waves and**
**wind more pronounced than other tests. When the chamber was shallowly placed in soil (red dots), most of the reference flow was**
**lost from leaks at the edges. The best performance was achieved when chamber was placed 18 cm below ground and with the edges**
**soaked with water (green inverted triangles).**


Public domain. CC0 1.0.



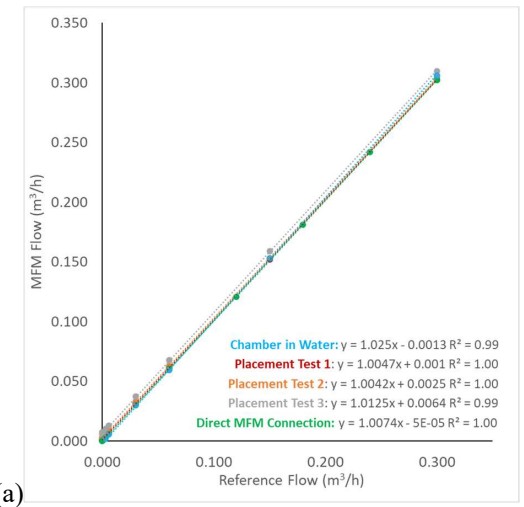

(a)

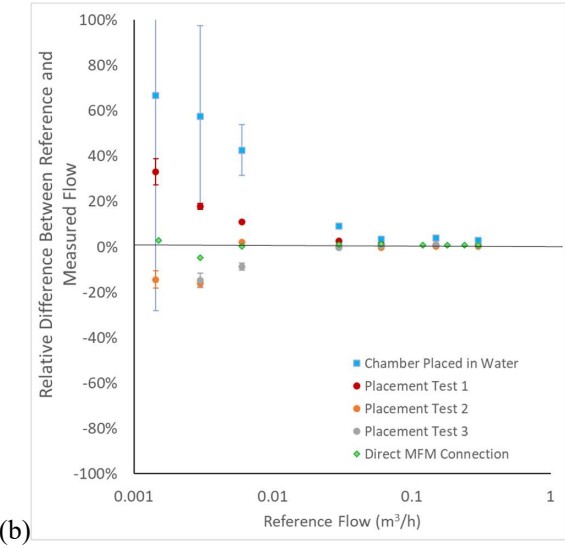

(b)

**Figure 7: Comparison of the references gas flow rates into the chamber and the measured flow rates out of the chamber mass flow meter (MFM) under the direction connection, chamber in water, and replicate (3x) 18 cm placement scenarios. (a) Measurements and linear regression of the reference and measured flows (b) The difference between the reference flow (MFC) and the measured flow (MFM) as a function of reference flow rate, showing that the error in flow rates becomes much greater below $1 \times 10^{-2}$ m$^3$/h.**




**Table 1: Comparison of performance specification of the mass flow meter (MFM) used in this study along with the results of laboratory cross-calibration by direct connection between the MFM and a mass flow controller (MFC), and results from tests with the MFM chamber placed in water or with the repeat tests edges buried 18 cm in the ground. Limit of Detection (LOD) was calculated as three times the standard deviation of the signal with no reference flow over a 1-min period. The average uncertainty was calculated as the average relative standard deviation (RSD) from the reference flows that were measured to be above LOD for each test and are expressed in percent. Also included is the mass-based limit of detection for a pure methane leak.**


| | | Measurement / Reference value | Intercept | Baseline Standard deviation | LOD | LOD | Average Uncertainty |
|---|---|---|---|---|---|---|---|
| | Units→ | $(m^3/h)/(m^3/h)$ | $(m^3/h)$ | $(m^3/h)$ | $(m^3/h)$ | $CH_4$ g/h | % RSD |
| MFM Specification | | 1.0000 ± 0.0075 | 0.0 | NA | 0.0006 | 0.43 | 0.10% |
| Direct MFM Connection | | 1.0074 | $5.0 \times 10^{-5}$ | 0.00251 | 0.00752 | 0.54 | 0.11% |
| Chamber Placed in Water | | 1.025 | $5.0 \times 10^{-3}$ | 0.00126 | 0.003786 | 2.71 | 20.90% |
| 18 cm Placement Test 1 | | 1.0047 | $1.0 \times 10^{-3}$ | 0.00346 | 0.001038 | 0.74 | 1.10% |
| 18 cm Placement Test 2 | | 1.0042 | $2.5 \times 10^{-3}$ | 0.00987 | 0.000818 | 0.58 | 0.81% |
| 18 cm Placement Test 3 | | 1.0125 | $6.4 \times 10^{-3}$ | 0.00462 | 0.001385 | 0.99 | 0.67% |

## 5.2 Laboratory Test Results: MFC-MFM Cross-Calibration, MFM Chamber Stability, and Detection Rate

Cross-calibration of the MFM and MFC showed a linear response and accurate cross-calibrations, with the slope of the MFM/MFC response being 1.0074 with an $R^2$ of 1.00. The flow resistance of the MFM over the reference flow range of 0 to 0.3 $m^3/h$ was 9.07 $10^{-4}$ atm/$(m^3/h)$, lower than the 2.72 $10^{-3}$ atm/$(m^3/h)$ at a flow rate of 6 $m^3/h$ specified by the manufacture. The ratio of the response of the MFM was 1.0075 of the output set on the reference MFC, with gas flow detectable down to 7.52 $10^{-3}$ $m^3/h$. This is essentially the same as the stated manufacture performance and demonstrates the best possible performance of the MFM in isolation from the chamber, vent, and environmental processes, serving as a baseline for subsequent tests (Fig. 7 and Table 1). The result of repeated flow cycles showed the response of the MFM chamber was stable, with a relative standard deviation in response of 0.35% over the 15-hour test window. The mean limit of detection (LOD), estimated from three times the standard deviation ($3\sigma$) of the zero-flow MFM signal of each measurement cycle was $1.02\times10^{-3} \pm 3.61\times10^{-4}$ $m^3/h$, corresponding to $0.74 \pm 0.26$ g/h of methane and 2.02 g/h ± 0.71 g/h of $CO_2$. Experimentally, the mean detection level (defined as the MFC flow that raises the MFM signal to above $3\sigma$ of the background level) was $9.12 \times10^{-4}$ $m^3/h$ ranging from $6.0 \times10^{-5}$ $m^3/h$ to $3.0 \times10^{-3}$ $m^3/h$. The system had a 100% detection rate for flows rates equivalent to 2.15 g/h of methane or less and 70% detection rate for flows equivalent to 1 g/h of pure methane or lower. The presence a range of detection levels in the repeated study indicates that the film seal base seal was not always rigid, and instead it expanded during the experiment, allowing several low flow stages to complete without MFM detection.

Public domain. CC0 1.0.

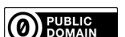



### 5.3 Gas Flow through Soil can Reduce MFM Chamber Accuracy

**5.3.1 Estimating the Effect of Soil Permeability**

The proportion of the gas flow through the MFM relative to the soil is dependent on the flow resistance of the MFM relative to the flow resistance of the soil under the chamber. Reviewing the specifications of MFM that are marketed for measuring gas leaks in upstream oil and gas operations and general-purpose MFM shows that the ratio of maximum flow to pressure drop at maximum flow varies from around 20 (m$^3$/h)/atm (most restrictive) to 14.4 ×10$^4$ (m$^3$/h)/atm (least restrictive), with three

examples in the range of 1 – 3 ×10$^3$ (m$^3$/h)/atm (Ventbuster Instruments, 2024; VentMEDIC Corporation, 2024; Calscan, 2024; Alicat Scientific, 2024a, b). Using MFMs that are restrictive may result in an underestimation of emissions as a substantial portion of the gas will leave the chamber through the soil under most conditions.

Surface soils have permeabilities that range from ≈10$^{-10}$ m$^2$ or more for unsaturated materials with large flow paths to 10$^{-13}$ and less for materials with constricted flow paths, water content, or fine particles like clay soils (Bloomfield and

Williams, 1995; ASTM International, 2013; Wałowski, 2017; Benavente et al., 2019). Assuming laminar flow (i.e. a linear relationship between pressure and flow) through the MFM and soil flow governed by Darcy's Law (equation 2), the ratio of gas flowing out of the MFM to gas flowing through the soil should not change with pressure at gas flow rates that are sufficiently low that turbulent flow does not occur in the MFM and the soil permeability does not change in response to the pressure. Figure 8 addresses the behaviour of different MFM soil combinations, including the fraction of air flow through the

MFM at permeabilities between 10$^{-13}$ to 10$^{-9}$ m$^2$ at idealized flow meter maximum flow pressure drop ratios of 20 to 20,000 (m$^3$/h)/atm in a chamber with a footprint area of 0.13 m$^2$ and a mean soil flow path of 0.06 m. Where soils have low permeability, 98% (or more) of the air flow is captured by the MFM, regardless of its flow resistance. As soil permeability increases, MFMs that have greater flow resistance will have less of the gas flow pass through them. At a permeability of 10$^{-11}$ m$^2$, (approximately dry sandy soil), an MFM with a pressure drop ratio of 20 (m$^3$/h)/atm (representative of low

flow/instrumentation targeted units) will have 34% of the air flow, while an MFM with a pressure drop ratio of 2,000 (m$^3$/h)/atm will still have 98%. The arrows in Fig. 8 illustrate how soil, gas, and chamber properties influence the amount of gas from a leak flowing through the MFM. The permeability of the soil is reduced by water saturation, so added water content will cause more flow through the MFM, and flow through the MFM will be reduced as soil dries. The dynamic viscosity of the gas in the leak will also affect the flow through the soil. Flows of gases with dynamic viscosities greater than air (non-

methane hydrocarbons, for example) will have greater flow through the MFM, while gases with lower viscosity (like carbon dioxide and methane) will have greater flow through the soil (Davidson, 1993; Huber and Harvey, 2011). Increasing the chamber area increases the flow paths through the soil, reducing the proportion of MFM flow. Increasing the chamber placement depth lengthens the gas flow paths through the soil, increasing the proportion of gas flow through the MFM. Overall, the lower the flow resistance of the MFM, the broader the range of soils and scenarios where the chamber can be placed without

significant gas flow losses through the soil.

Public domain. CC0 1.0.

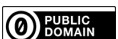



While the ratio of the flow between the MFM and the soil does not depend on pressure, the magnitude of the flow through the MFM and the soil does. Figure 9 demonstrates the quanity of air flowing through soil of different permeability as pressure in the chamber increases over the dynamic range of a low resistance MFM (0 – 30 m$^3$/h, 14,444 (m$^3$/h)/atm) on a chamber with the same configuration as Fig. 8. At the maximum MFM flow and pressure drop ($2 \times 10^{-3}$ atm), high permeability

soils could be passing several cubic meters per hour of gas that is not captured by the MFM. While this could be a small fraction of the total flow of the system, it may represent a significant underestimate of the magnitude of the leak in question. Following Fig. 8, arrows on the Fig. 9 indicate the effects of soil water, gas viscosity, and chamber characteristics on soil gas flow.

Public domain. CC0 1.0.




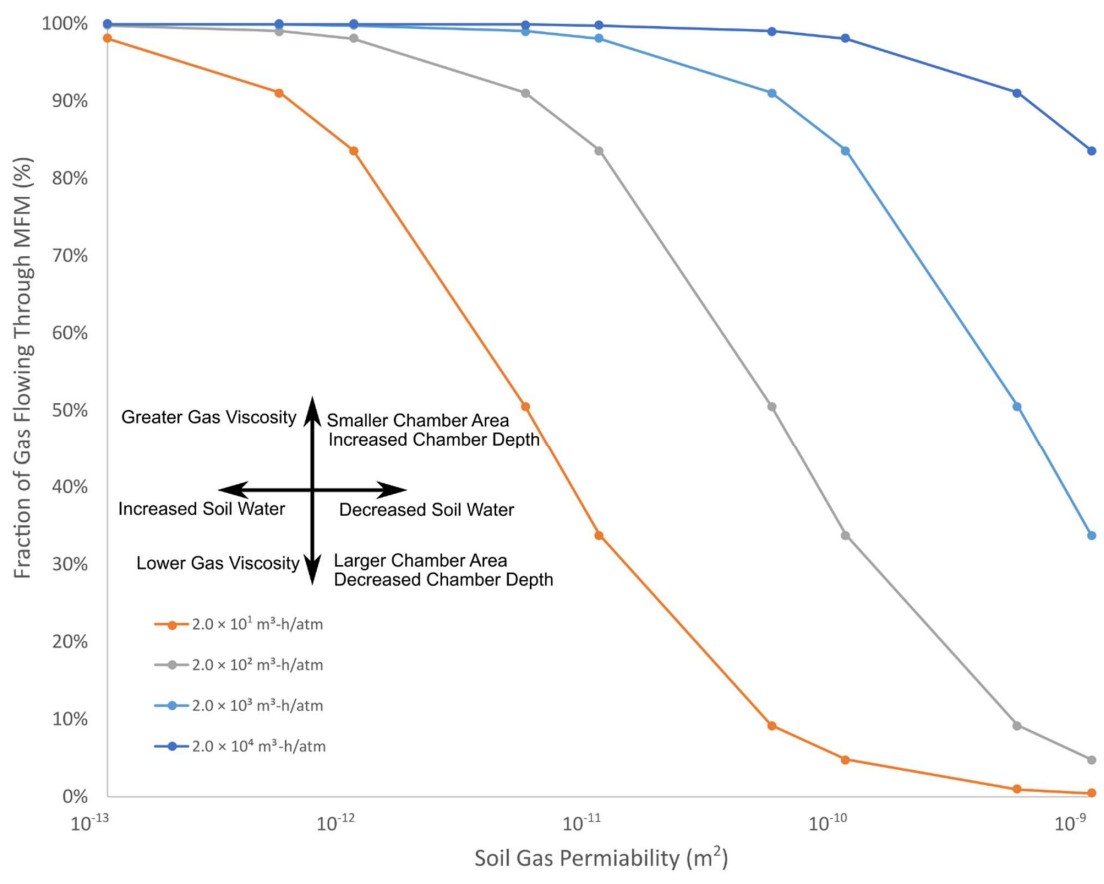

**Figure 8: Air flow through the mass flow meter (MFM) under the MFM chamber for different MFM flow resistances and differing soil gas permeability ranges for a chamber area of 0.13 m². The arrows demonstrate how the effects of soil water content, gas viscosity, and chamber properties affect the proportion of flow between the MFM and the soil.**




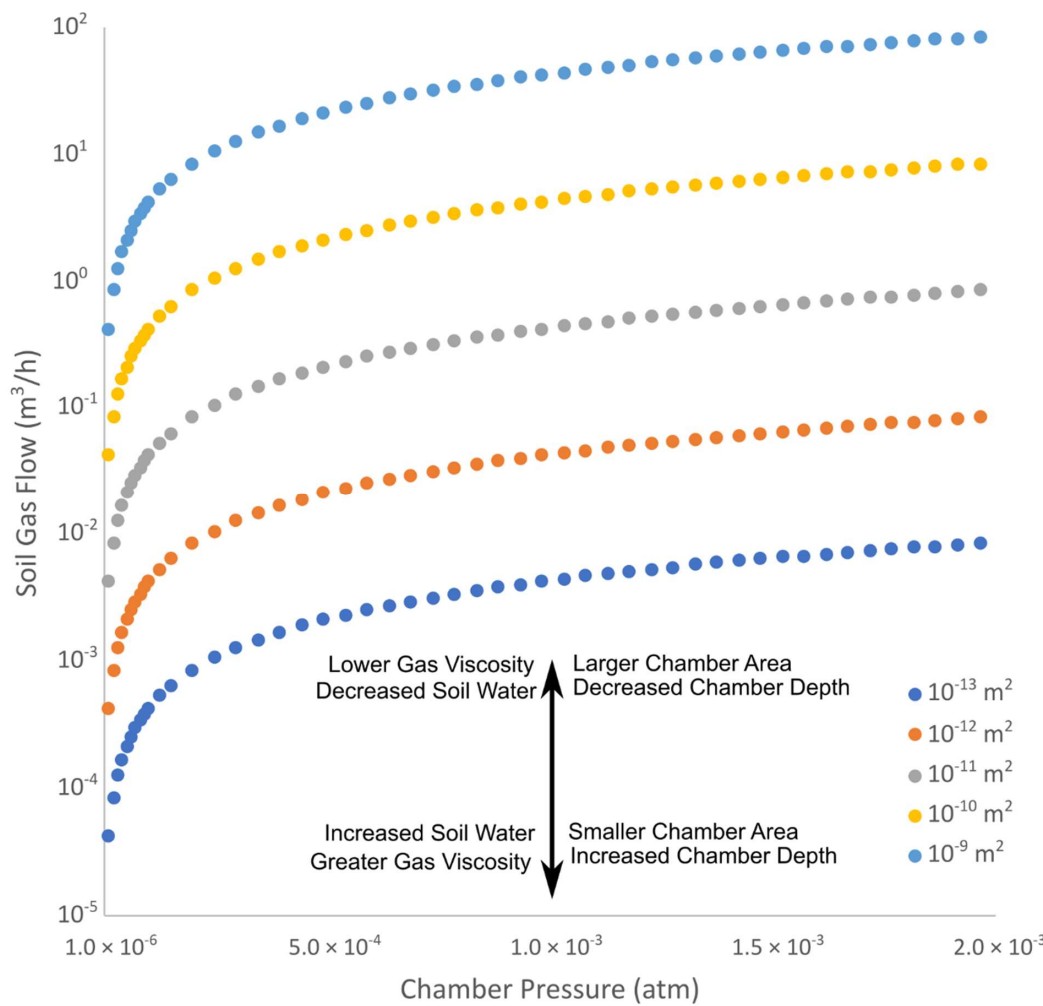

**Figure 9: Gas flow through different soil permeabilities ($10^{-9}$ to $10^{-13}$ m$^2$) as a function of chamber pressure. The arrows show how changes in conditions affect soil gas flow at a given chamber pressure.**


Public domain. CC0 1.0.

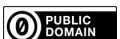



### 5.3.2 Environmental Processes Affect Gas Flow in the MFM Chamber

Environmental flow is a MFM chamber signal response that is the product of natural processes around the chamber and is not related to the source of the leak in question. The effects of various types of environmental flow are illustrated by the data collected from the MFC chamber in a range of conditions without an active gas source inside it (Fig. 10a). During the 11-day measurement period, the chamber experienced both diurnal cycles and several different weather conditions (sun/clouds/rain/wind) that created gas flow and signal changes in the chamber MFC. The 1-minute average MFC mass flow was -2.94 ×$10^{-3}$ standard $m^3$/h with a standard deviation of 7.40 ×$10^{-3}$ standard $m^3$/h. The highest 1-minute average mass flow was 3.21 ×$10^{-2}$ $m^3$/h. In terms of a methane emissions rate, this corresponds to a mean rate of -2.10 g/h, with a standard deviation of 5.29 g/h and a maximum of 22 g/h.

Environmental flows were observed to occur from many sources. Stack flow from temperature differences between the chamber and the outside air became noticeable when the differences were larger than approximately 3 ℃. The largest environmental flows and the periods of the greatest variability in MFC data were observed during periods when the average wind speed was greater than 10 km/h, though these were masked by the hourly average obtained from National Oceanic and Atmospheric Administration (NOAA) that does not identify instantaneous gusts that can briefly cause high MFC flows. The MFC flow relationship that was most difficult to identify was that of water vapor pressure in the chamber, due to its close correlation with thermal processes, however, water vapor movement through the chamber MFC or changing gas composition in the chamber contributed to small, slowly changing baseline flow signal drift in otherwise calm conditions.

To determine the contribution from different environmental processes to environmental flow, the measured MFC flow was fit to an orthogonal least squares (OLS) regression (Seabold, 2010) assuming a linear relationship among environmental flow ($Q_{env}$, $m^3$/h) and each variable (average wind speed ($W$, km/h), temperature difference ($\Delta T$, ℃), and water vapor pressure ($P_w$, atm), yielding the following equation to simulate the environmental flow in the MFM chamber system:

$$Q_{env}= 1.45×10^{-4} × W + 9.43 × 10^{-4} × \Delta T + (-7.55× 10^{-1}-) × P_w + 2.48 × 10^{-3} \qquad (4)$$

The estimated environmental flow is shown in Fig 10 a and b (blue line) along with the measured flow. The inability to obtain high time resolution wind speed data is a limitation on the utility of using a multi-parameter fit to examine the various sources of environmental flows in this study. There is considerable variation in the MFC measurements during periods of high wind and the associated instantaneous variation in windspeed that are not captured in the estimated environmental flow. As a result, the $R^2$ of the OLS fit was 0.36, and it is expected that higher frequency wind measurements, which are not typically available during routine emissions measurements, would improve the quality of the fit. The mean simulated environmental flow was -7.09×$10^{-4}$ $m^3$/h, with a standard deviation of 2.45 ×$10^{-3}$ $m^3$/h, and a maximum of 8.61 × $10^{-3}$ $m^3$/h, which is both smaller and less variable than the measured MFC flow.

To examine how each source of environmental flow contributes to the predicted flow over time, the contribution of each source (wind, temperature difference, and water vapor pressure) to the predicted chamber flow during the study is shown in Fig. 10 b. The percentage of the predicted flow from each source is shown in Fig. 10 c. Temperature-driven flow is only

Public domain. CC0 1.0.

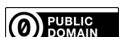



present during the daytime when the chamber can be heated by sunlight, while wind-driven flow and signals from water vapor

pressure can be present at any point in time.

Public domain. CC0 1.0.

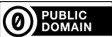



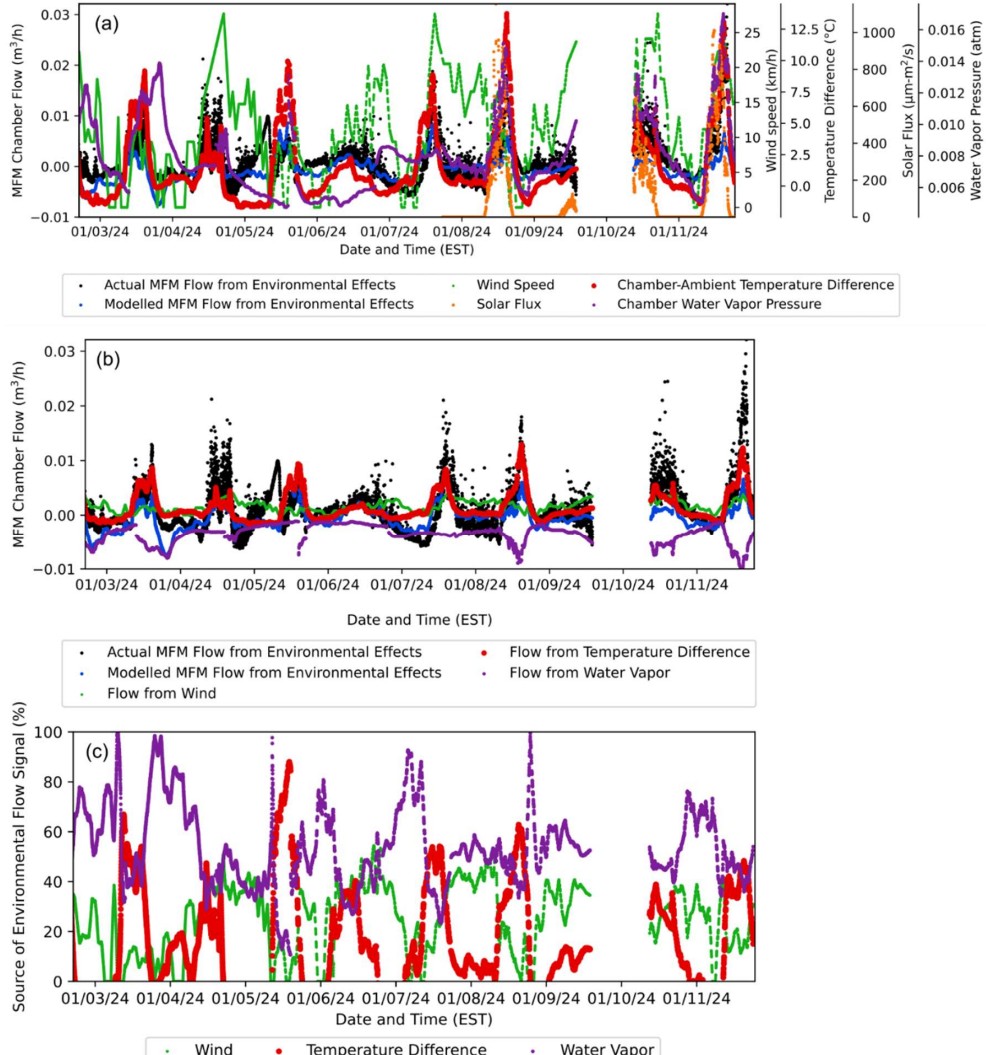

**Figure 10: Analysis of the gas flow through the mass flow meter (MFM) chamber from environmental processes (a) Test chamber environmental flow measurements along with wind speed, temperature difference, solar flux, water vapor pressure, and modelled environmental flow. (b) The measured and predicted environmental flow along with the contribution to the MFM signal from wind, temperature difference, and water vapor pressure. (c) The relative proportion of the environmental flow from each factor (wind, temperature difference, and water vapor pressure) calculated at the time of each measurement. The collected on 1/10/2024 are omitted because the humidity sensor stopped working.**





The role that water vapor plays in the flow rate reported by the MFC is complex. The baseline signal of the measured flow follows the water vapor pressure contribution (Fig. 10 a), indicating the influence of water vapor composition (rather than water vapor movement) on the baseline MFC signal (for example, around 00:00 eastern standard time (EST) on 1/7/2024). Increasing temperatures can also drive evaporation of water from soil, increasing the partial pressure of water vapor in the

chamber during periods where the largest positive flows from solar gain warming the chamber also had increases in negative apparent contribution from water vapor. Finally, unlike wind and temperature differences, there is always water vapor present in the air, and its partial pressure evolves more slowly than the other environmental flow sources, so it tends to dominate in the decomposition (Fig. 10 c). It is important to point out that all three sources of environmental flow are of similar magnitude during the period of the study.

420            Figure 11 contains a set of violin plots exploring the sensitivity of the environmental flow from the chamber to the different sources and the distribution of the different flows, along with a bar chart demonstrating how the different extremes of environmental sources can combine to create flow in the chamber. The measured environmental flow rates (black) have greater extremes than the modelled flow rates (blue), again illustrating that the hourly mean winds (green) likely lead to underprediction of wind driven flow. Thus, the temperature driven flow shows the most variability, including small negative

flows when the chamber is colder than the air. Both the observed and modelled environmental flows had means that were very near zero, demonstrating that there were considerable periods of time where there was no air movement through chamber and therefore, if the chamber had been placed over a leak, the reported flow by the MFM would be very close to the true value of the leak. A potential use of developing the modelled relationship for a MFM chamber system like equation (4) could be to isolate emissions measurements during intervals of time that have minimal influence from environmental sources.

430            We examined a set of environmental extremes for temperature, wind, and humidity to investigate the upper limits of environmental flows that could occur when using the MFM chamber to measure fluxes in the field. Temperature differences between ambient air and walls of a black chamber in direct sun might potentially range as much as 0–40 °C (Russell and Bartels, 1989; Chestovich et al., 2022), water saturated vapor pressures 0.000 to 0.0395 atm cover a range of extremely dry to humid areas (Brasseur et al., 1999), while wind speeds ranging from 0 to 50 km/h are used to capture a range over which a

large chamber could be easily anchored to the ground. Environmental flow from temperature difference could reach $3.8 \times 10^{-2}$ $m^3/h$ (27.0 g/h $CH_4$, 74.0 g/h $CO_2$) at 40 °C, average windspeeds of 50 km/h produce flows of $5 \times 10^{-3}$ $m^3/h$ (5.2 g/h $CH_4$, 14.3 g/h $CO_2$), and the negative relationship between water vapor pressure and measured MFM flow would create a signal of $-2.9$ $\times 10^{-2}$ $m^3/h$ ($-21.3$ $CH_4$ g/h, $-58.5$ $CO_2$ g/h). These extremes are notably larger than 25% and 75% quantiles of the measured and fit MFC data which are on order of $7 \times 10^{-3}$ $m^3/h$ (5 g/h $CH_4$, 14 g/h $CO_2$) or less during the period of the experiment.

Public domain. CC0 1.0.



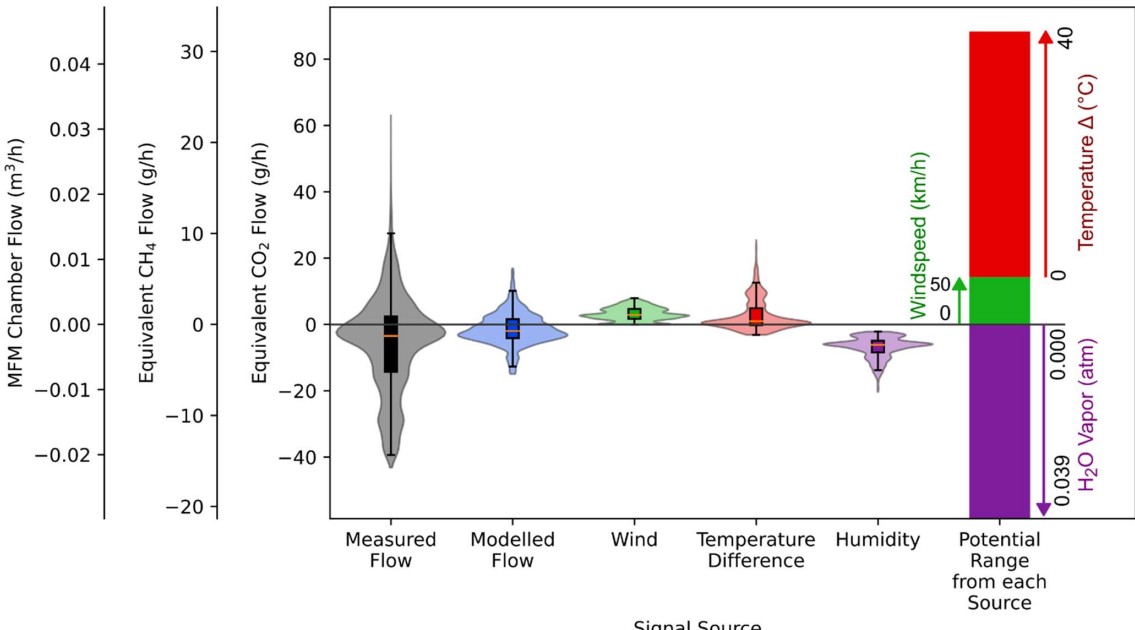


**Figure 11: Violin and box plots showing the range and distribution of measured chamber flow (black) with modelled chamber flow from equation 4 (blue), and contributions to the modelled flow from wind (green), temperature difference between inside the chamber and outside (red), and humidity of the gas at the mass flow meter (MFM) vent (purple). On the right, the range of potential contributions for the different environmental components is shown as stacked bars with arrows showing the potential contribution from each environmental factor. Temperature differences and wind increase measured flow, while increasing water vapor in the chamber causes the signal in the MFM to decrease. The left axis is plotted both in terms of volumetric gas flow through the MFM (corrected to 25 °C and 1 atmosphere) as well as emission rates of methane and carbon dioxide.**

This test demonstrates that uncertainty from environmental flow in the MFM chamber can be reduced by limiting the temperature difference of the chamber (for example, choosing reflective materials and not measuring in direct sunlight), and minimizing measurements during windy conditions. While it is conceptually desirable to limit evolution of water vapor in the chamber, this is likely not practicable as both the soil and likely the emissions source have some amount of water vapor in their composition, and thus the influence of water vapor is unavoidable, though different MFCs may have different response factors to water vapor. Another benefit of insuring a good seal against the ground is that environmental flows could be reduced as conductance through soil is much lower than from leaks around the chamber edges. Understanding the factors that cause environmental flow is also important because these processes may also mediate the emissions rates of the leak targeted for measurement. Last, selecting periods of time when the factors are not major contributors to the flow from the chamber could improve the accuracy of the measurement, especially for small leaks (Maier et al., 2019; Forde et al., 2019; Fleming et al., 2021). This is particularly a concern for smaller leaks on the order of 0.015 m³/h (~10 g/h CH₄, 30 g/h CO₂). For example,



open hole orphan wells may themselves be modulated by environmental processes (Riddick et al., 2020) as their magnitude
and variation could be masked by environmental flow through the chamber.

### 5.4 Comparing MFM Chamber and Dynamic Chamber Response

MFMs have different response factors for gases of different compositions (Tison, 1996; Hardy et al., 1999; MKS Instruments,
2024; Alicat Scientific, 2014). The composition of the gas in the MFM chamber changes during the measurement period as
the air in the chamber is displaced by the gas emitted from the source. Figure 12 a, c, e shows the evolution of the gas in the
MFM and dynamic chambers under $1.40\times10^{-3}$ m$^3$/h, $6\times10^{-2}$ m$^3$/h, and pulsed $6\times10^{-2}$ m$^3$/h  (1 g/h and 42.9 g/h CH$_4$) scenarios.
Figure 12 b, d, f shows the corresponding simulated measurements from both systems in comparison with the actual emission
rate in the chambers, using a response factor for methane in the MFM chamber (CO$_2$ would show a similar behavior, but with
a different MFM correction factor).

For the slowest leak (Fig. 12 a and b), the MFM chamber does not fully saturate with methane, reaching a peak mixing
ratio of 0.125. This results in an over-estimation of the leak rate throughout the emissions measurement period, which could
be corrected by measurements of gas concentrations in the MFM chamber. In contrast, the dynamic chamber reaches
concentration equilibrium within a few hours, and the measured emissions rate matches the true emissions rate throughout. At
a moderate leak rate (Fig. 12 (c) and 12 (d)), the MFM chamber becomes fully saturated with methane gas after 20 h and the
measured emissions rate during the last 4 h match the actual leak rate. It is notable that the dynamic chamber has the same
equilibration time in both scenarios. Figure 12e shows that episodic emissions of methane from pulsed flow (for example, from
bubbles rising through a wellbore) can create a situation where the concentration of methane in both chambers do not reach
equilibrium over the course of the measurement period. In the MFM chamber, the concentration increases in a step function
as each pulse accumulates in the chamber. In the dynamic chamber, which is continually removing gas, the concentration
spikes, then decays with each pulse, never reaching equilibrium. In Figure 12 f shows the emission rate measurements under
pulsed conditions: the MFM chamber is able to capture the timing of the emissions pulses accurately, though it over-reports
the emission rate due to the mismatch between the methane response and the actual gas composition in the chamber. This
discrepancy decreases as the concentration of methane in the chamber increases. In contrast, the dynamic chamber captures
the average emissions rate, but the amplitude and timing of pulses are damped relative to the true value.

Public domain. CC0 1.0.



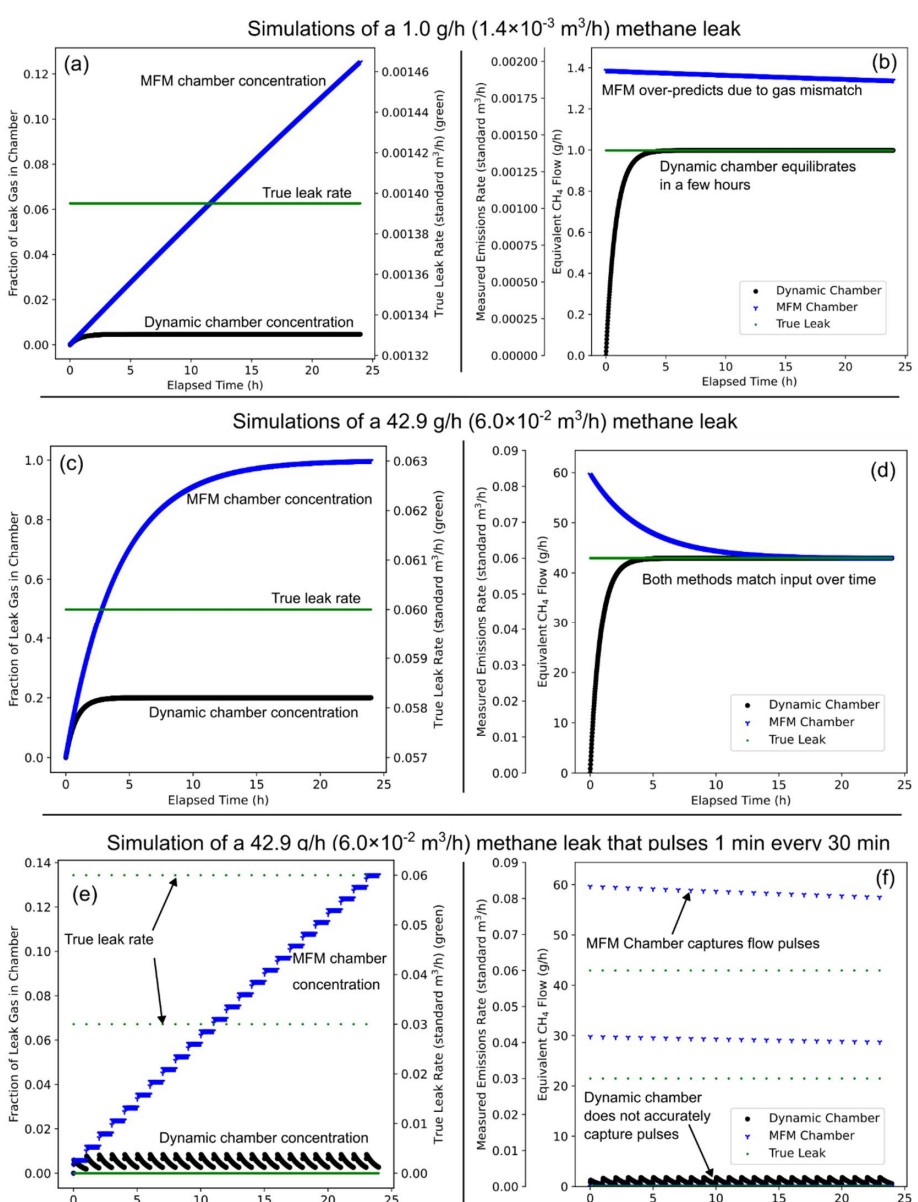

**Figure 12: Simulations of mass flow meter (MFM) and dynamic chambers with different size leaks and pulsed flows that simulate intermitted leaks. The simulated chambers volumes are 250 L and the simulated dynamic chamber pump rate was 5.0 L/min. Left column (a, c, and e): Simulated fraction of leak gas in MFM and dynamic chambers. The leak rate measurements that would be reported by the MFM and dynamic chamber systems are shown in the right column figs. (b, d, and f).**

Public domain. CC0 1.0.



These simulations highlight several characteristics of MFM chambers relative to measurements from dynamic chambers and direct MFM connections to emission sources. First, the MFM chamber volume can act as reservoir that must be flushed before the response of the MFM to the source gas is stable. For measurements of small leaks, the purge period before a stable response can be lengthy, so multiple concentration measurements may be required to correct the MFM measurements over the data collection period. This purge interval for a direct connection or the dynamic chamber is much smaller, and in the

case of the dynamic chamber, not dependent on the emission rate. The MFM chamber is also able to resolve instantaneous flows like an MFM directly connected to the leak source, while the dynamic chamber measurement technique shows a damped response as the change in emission rate is resolved as a change in concentration rather than in flow, though it is still able to capture the average emissions over time. Notably, with information about the volume of the chamber and the concentration of balance gas in the chamber, it is possible use the derivative of the dynamic chamber concentration to directly capture the

variation in emissions rate (Gao and Yates, 1998). However, we are not aware of this being applied to measurements of emissions from orphan wells or included in other chamber guidelines (Maier et al.; Liu et al., 2022; Williams et al., 2023; Riddick et al., 2023). In any case, the MFM based approaches can quickly resolve changes in emissions rate without the need for higher order corrections and do so without the need for pumps and/or mixing of the chamber contents with a fan, though at lower emission rates the period required to purge the chamber with gas could be very long, resulting in inaccuracy without

detailed concentration corrections.

**6 Suggestions for Optimal Emission Measurements Using the MFM Chamber**

To optimize emission measurements using the MFM Chamber:

(1)     Reduce the chamber size to a minimum to reduce soil footprint and purge volume and select an MFM with minimal flow restriction can help reduce uncertainties associated with the method.

(2)     The edges of the rigid chamber need to be sealed against the ground through a combination of deep burial (7 cm or more in our findings), soaking the soil with water to encourage settling and reducing permeability, and adding weights on top of the chamber or other measures to prevent the chamber from shifting and breaking the soil seal.

(3)     When setting up the MFM chamber over an active leak, the flow rate of gas can be monitored while sealing the edge of the chamber into the soil to achieve a maximum flow and identify when the chamber is correctly placed.

For wells and sources that are located on concrete, gravel, or other high gas permeability surfaces where a good seal is difficult to create, it might be possible to pour a bentonite or other impermeable mud slurry to seal the area around the leak and the base of the chamber. However, considerations for how this affects humidity in the chamber might be required and the slurry could block below-ground leaks from being detected.

(4)     For highest accuracy, measurements should be made during low wind periods where the chamber is not directly

in the sun (shaded location, overcast, or nighttime). Extreme weather conditions can create environmental flows



that can increase the uncertainty or completely obfuscate the measurements from small leaks, especially below 0.04 $m^3$/h (~40 g/h $CH_4$, ~90 g/h $CO_2$).

(5)    For measurements of small and episodic sources, the change in gas concentration in the chamber could be accounted for either by placing the chamber on the source for a sufficiently long period to allow it to fill with

525          gas or by making periodic concentration measurements over the measurement interval, instead of a single measurement at the end of the measurement session.

(6)    Document the reference conditions (temperature and pressure) as well as the gas for which the MFM is calibrated to increase clarity for data processing and reporting.

(7)    High data collection frequency from the MFM (>1 $s^{-1}$) can help identify both sources of environmental flow as

530          well as short duration changes in emission rate.

## 7 Conclusion

The critical parameters describing the performance of the MFM chamber tested in this study are summarized in Table 2. Field testing of the MFM chamber shows it can quantify leaks as small as 9.12 $10^{-4}$ $m^3$/h air, corresponding 0.65 g/h $CH_4$ or 1.78

g/h $CO_2$. The quality of the ground seal of the chamber is critical for obtaining high quality data. The chamber body, all access ports and fittings, and its ground seal need to be leak free. The effect of gas flow through soil on the accuracy of the measurements is minimized by selecting a MFM with the lowest pressure drop possible. A high-quality ground seal can be made by deeply burying the chamber edges (15 cm in our study), sealing the edges with soil free of roots and rocks that increase permeability, and wetting and packing the soil to create a tight seal. The deeper the chamber is placed, the more reliable the

seal will be. Gas permeability of the soil around the MFM chamber can cause bias in measurements. Soil with high gas permeability (for example, dry, gravelly soil) will provide a gas flow path that reduces the flow through the mass flow meter. The effect of soil permeability can be reduced by using the lowest flow resistance meter possible and minimizing the footprint of the chamber on the ground. The measurements made by the chamber are influenced by wind, temperature differences between the chamber and the atmosphere, and variation in humidity during measurements. These influences can contribute to

the leak signal measured by the MFM, potentially ranging from -2 to $4\times10^{-2}$ $m^3$/h, though in practice, these sources contribute less than $\pm$ $1\times10^{-2}$ $m^3$ ( $\approx$ $\pm$ 7 g/h $CH_4$ and $\approx$ $\pm$ 20 g/h $CO_2$). The MFM chamber may be more appropriate than dynamic chambers for time series measurements owing to the instantaneous response and simpler implementation.

Public domain. CC0 1.0.

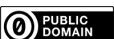



**Table 2: A summary of the critical parameters tested in this study in terms of mass air flow, methane mass emission rate, and carbon dioxide maximum emission rate, along with potential contributions to uncertainty from environmentally derived signals in the MFM chamber. BG: Background. Eq. 8: corresponds to values calculated using equation 8 at the noted inputs.**

| Parameter | LOD | Mean Detection Level | 100% Detection Level | Average Uncertainty | Observed Environ. Flow | Max Wind Flow | Max Solar Flow | Max Humidity Signal |
|---|---|---|---|---|---|---|---|---|
| Method | $3\sigma$ of BG | $3\sigma$ over BG | $3\sigma$ over BG | RSD of all levels (%) | $1\sigma$ of 11 days | Eq. 8 @ 50 km/h | Eq. 8 @ 40℃ | Eq. 8 @ 0.0390 atm $H_2O$ |
| Air $m^3$/h (STP) | $1.08\times10^{-3}$ | $9.12\times10^{-4}$ | $3.00\times10^{-3}$ | 0.86% | $7.40\times10^{-3}$ | $5.00\times10^{-3}$ | $3.80\times10^{-2}$ | $-2.90\times10^{-3}$ |
| Pure $CH_4$ g/h | $7.70\times10^{-1}$ | $6.50\times10^{-1}$ | 2.14 | 0.86% | 5.27 | 5.2 | 27 | -21.3 |
| Pure $CO_2$ g/h | 2.11 | 1.78 | 5.87 | 0.86% | 14.48 | 14.3 | 74 | -58.5 |

The results of this study also reveal several opportunities for future refinement of the methodology. Wind driven variation could be reduced by using an engineered chamber vent that reduces venturi driven flow (Xu et al., 2006). The MFMs could be built to include $CH_4$, $CO_2$, $O_2$, humidity, and other common industrial sensors to measure the concentration of the gases flowing through the MFMs. Measuring the gases in this way could have several benefits: the concentrations of gases can be very high, in a range where low cost industrial and HVAC targeted sensors perform well, allowing for real time speciation of emissions and correction for gas sensitivity of the MFM, improving accuracy and reducing the need for offline sample analysis. The testing methods used in this study also offer the opportunity to calibrate/validate MFM deployments in the field by adding known compressed air flows to chambers measuring active leaks, though consideration is required given the low-pressure differentials present in the system, as adding additional gas flow to the chamber could suppress the leak rate or force air back into the leak source.

**Data availability**

All the data collected in the study are publicly available on the internet as a U.S. Geological Survey (USGS) Data Release (Haase and Gianoutsos, 2025).

Public domain. CC0 1.0.

**Author contributions**

KBH: Designed and executed the study including all laboratory and field tests, managed the data, performed the analysis, and
led the construction of the manuscript.

NJG: Contributed to study conceptualization, reviewed and supported study execution reviewing and providing critical
feedback on results, and contributed manuscript writing and overall scope of the product.

**Acknowledgements**

The authors wish to thank: Curtis Shuck at the Well-Done Foundation for first introducing us to the MFM Chamber approach.
Robert Layher at Ventbuster Instruments and Michael Beck of Vent Medic for detailed information and insights about the data
format and flow properties of MFMs. The DOI Methane Interagency Team, especially Jeff Sorkin, Linda Gieser, and Jacob
Deal (U.S. Department of Agriculture Forest Service) for helping us understand aspects of the MFM chamber that could be
explored. Aimee Haase who provided the soil plots for testing, and Natalie Pekney and Matthew Reeder of the National Energy
and Technology Laboratory for their helpful insights into the challenges and potential benefits of MFM chambers for orphan
well emissions measurements.

**Disclaimer**

*Any use of trade, firm, or product names is for descriptive purposes only and does not imply endorsement by the U.S.
Government.*

**Funding**

This project was supported by the U.S. Department of the Interior Orphaned Wells Program Office, as part of the
implementation of the Infrastructure Investment and Jobs Act (IIJA), also known as the Bipartisan Infrastructure Law (BIL).

**Competing interests**

The USGS has a collaborative research agreement with the Well-Done Foundation.
This research was done independently by the USGS, and the authors declare there is no conflict of interest.



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

Methods Used to Measure Methane Emissions of 1 g CH4 h−1, Sensors, 23, 9246, 2023.

Riddick, S. N., Mbua, M., Santos, A., Emerson, E. W., Cheptonui, F., Houlihan, C., Hodshire, A. L., Anand, A., Hartzell, W.,
and Zimmerle, D. J.: Methane emissions from abandoned oil and gas wells in Colorado, Science of The Total Environment,
922, 170990, doi: 10.1016/j.scitotenv.2024.170990, 2024.

Russell, D. G., and Bartels, R. A.: The temperature of various surfaces exposed to solar radiation: An experiment, The
Physics Teacher, 27, 179-181, doi: 10.1119/1.2342710, 1989.

Saint-Vincent, P. M. B., Reeder, M. D., Sams III, J. I., and Pekney, N. J.: An Analysis of Abandoned Oil Well Characteristics
Affecting Methane Emissions Estimates in the Cherokee Platform in Eastern Oklahoma, Geophys. Res. Lett., 47,
e2020GL089663, doi: 10.1029/2020GL089663, 2020a.

Saint-Vincent, P. M. B., Sams, J. I., III, Hammack, R. W., Veloski, G. A., and Pekney, N. J.: Identifying Abandoned Well
Sites Using Database Records and Aeromagnetic Surveys, Environ. Sci. Technol., 54, 8300-8309, doi:
10.1021/acs.est.0c00044, 2020b.

Seabold, S. a. J. P.: Statsmodels: Econometric and Statistical Modeling with Python, Proceedings of the 9th Python in
Science Conference, 2010.

Thalasso, F., Riquelme, B., Gómez, A., Mackenzie, R., Aguirre, F. J., Hoyos-Santillan, J., Rozzi, R., and Sepulveda-Jauregui,
A.: Technical note: Skirt chamber – an open dynamic method for the rapid and minimally intrusive measurement of
greenhouse gas emissions from peatlands, Biogeosci., 20, 3737-3749, doi: 10.5194/bg-20-3737-2023, 2023.

Tison, S. A.: A critical evaluation of thermal mass flow meters, Journal of Vacuum Science & Technology A, 14, 2582-2591,
doi: 10.1116/1.579985, 1996.

Townsend-Small, A., Ferrara, T. W., Lyon, D. R., Fries, A. E., and Lamb, B. K.: Emissions of coalbed and natural gas
methane from abandoned oil and gas wells in the United States, Geophys. Res. Lett., 43, 2283-2290, doi:
10.1002/2015GL067623, 2016.

Townsend-Small, A., and Hoschouer, J.: Direct measurements from shut-in and other abandoned wells in the Permian Basin
of Texas indicate some wells are a major source of methane emissions and produced water, Environmental Research Letters,
16, 054081, doi: 10.1088/1748-9326/abf06f, 2021.

U.S. Environmental Protection Agency: Methane Emissions from Abandoned Coal Mines in the United States: Emission
Inventory Methodology and 1990-2002 Emissions EstimatesEPA 430-R-21-021, 90, 2004.

The Ventbuster: https://www.ventbusters.com/the-ventbuster, access: 4/9/2024, 2024.

The Ventmedic: https://ventmedic.com/, access: 4/9/2024, 2024.



Vogt, J., Laforest, J., Argento, M., Kennedy, S., Bourlon, E., Lavoie, M., and Risk, D.: Active and inactive oil and gas sites contribute to methane emissions in western Saskatchewan, Canada, Elementa: Science of the Anthropocene, 10, 00014, doi: 10.1525/elementa.2022.00014, 2022.

Wałowski, G.: Assessment of gas permeability coefficient of porous materials, Journal of Sustainable Mining, 16, 55-65, doi: 10.1016/j.jsm.2017.08.001, 2017.

Williams, J. P., Risk, D., Marshall, A., Nickerson, N., Martell, A., Creelman, C., Grace, M., and Wach, G.: Methane emissions from abandoned coal and oil and gas developments in New Brunswick and Nova Scotia, Environmental Monitoring and Assessment, 191, 479, doi: 10.1007/s10661-019-7602-1, 2019.

Williams, J. P., Regehr, A., and Kang, M.: Methane Emissions from Abandoned Oil and Gas Wells in Canada and the United
States, Environ. Sci. Technol., 55, 563-570, doi: 10.1021/acs.est.0c04265, 2021.

Williams, J. P., El Hachem, K., and Kang, M.: Controlled-release testing of the static chamber methodology for direct measurements of methane emissions, Atmos. Meas. Tech., 16, 3421-3435, doi: 10.5194/amt-16-3421-2023, 2023.

Xu, L., Furtaw, M. D., Madsen, R. A., Garcia, R. L., Anderson, D. J., and McDermitt, D. K.: On maintaining pressure equilibrium between a soil $CO_2$ flux chamber and the ambient air, Journal of Geophysical Research: Atmospheres, 111, doi:
10.1029/2005JD006435, 2006.

Yang, D., Wang, W., Chen, W., Tan, X., and Wang, L.: Revisiting the methods for gas permeability measurement in tight porous medium, Journal of Rock Mechanics and Geotechnical Engineering, 11, 263-276, doi: 10.1016/j.jrmge.2018.08.012, 2019.

Yang, X., Lu, C., Huang, X., and Luo, J.: A rainfall event may produce a biased estimation of the indoor vapor intrusion risk
through exterior soil-gas sampling, Journal of Hydrology, 603, 127117, doi: https://doi.org/10.1016/j.jhydrol.2021.127117, 2021.

Zer0six: Offset-Emissions: https://www.zerosix.co/offset-emissions/, access: 4/12/2024, 2024.

Zhan, L.-t., Qiu, Q.-w., Xu, W.-j., and Chen, Y.-m.: Field measurement of gas permeability of compacted loess used as an earthen final cover for a municipal solid waste landfill, Journal of Zhejiang University-SCIENCE A, 17, 541-552, doi:
10.1631/jzus.A1600245, 2016.