# Peer review of "Evaluating mass flow meter measurements from chambers for greenhouse gas emissions from orphan wells and other point sources"

_EGUsphere, 2025_

## Author Response (AR1)

**RC1**: 'Comment on egusphere-2025-1201', Anonymous Referee #1, 17 Jun 2025

Overall Comments:

This study provides insights on the best practices for measuring emissions from orphaned wells and other small, ground-level static point sources using a simple mass flow meter (MFM) based measurement system. The chamber test methods that the authors used are sound and directly applicable to real-world scenarios. The importance of this research is reflected in public and private initiatives to plug orphaned wells, and this work focuses on a resource-efficient method for determining emissions from orphaned wells. As orphan well emissions have been shown to vary over time, these results are especially relevant for firms validating carbon credits and municipalities verifying well plug effectiveness as the MFM method does not require constant supervision and is simple in design. There are some comments to address prior to acceptance, though I believe that the results of this paper are important to disseminate for future orphan well research.

> KBH: Thank you for the interest and support for the work. We understand that it takes time and effort to provide constructive critical reviews, and we appreciate the efforts you have put in. We have modified the manuscript to account for your input and observations and believe it to be more complete and clear in the revised form.

Major Revisions:

(Line 195) "... sequentially decreased through 9 steps to 6.0 x 10^(-5) m^3/h ..." - This lowest step before zero is below the manufacturer's (Alicat) advertised accuracy of full scale. Please state clearly the uncertainties related to MFC and MFM measurements, and the combined MFC/MFM measurement uncertainties relevant for cross calibration results in Section 5.

> KBH: In the revised manuscript we have clarified that the accuracy of the MFM and MFC were each 0.1% of the full scale flow rate, with a combined accuracy of 0.14%, and that the purpose of going below the stated accuracy of the MFM was to capture the onset of the MFM signal.
>
> The purpose of testing the range of gas flow rates below the MFM response was to resolve the onset of MFM signal and any non-linearity at detected below-specification flows. While we found in our testing that the 100 slpm MFM did not resolve lower flows below the MFCs full scale manufactures full-scale range and was linear (Table 1 in the manuscript), we continued to use the MFM/MFC control script to simply the use of our data post processing templates.

(Eq. 4) - How are we sure that these measured variables are required/important in the linear model? A traditional regression analysis with t-tests on regression coefficients would suffice if put into the SI. The time series of these variables in Fig. 10 were fine, though x-y plots of environmental

variables versus actual MFM flow would be clearer.

> KBH: We originally omitted the t-test results in the manuscript as, at the time, it was apparent through direct observation that these factors were causing flow signal in the MFM. We realize that the reader needs more information since they were not present. Other factors could be at work beyond wind, temperature, and humidity, and there are many limitations to using a linear model to parameterize the environmental flow in the chamber. For example, as we note below, water vapor pressure and temperature are coupled values and there are likely other ways to address the flow caused by these variables. We have clarified in the revised manuscript that the P-values for all the coefficients was less than 0.005 and included a table of t-values, P-values, standard errors, and confidence intervals of the fit used in equation 4 in the SI.

(Line 390) "… considerable variation in the MFC measurements during periods of high wind …" - Further investigation should be done on variability of MFC measurement versus wind speed. This result is important for the feasibility of MFM measurements in high-wind areas. Suggestions for possible avenues of investigation include 1-hr std. dev. vs wind speed, difference of NOAA wind speed and gusts versus MFC std. dev., refitting Eq. 4 with standardized variables.

> KBH: To further evaluate the variability caused by wind, we have re-examined the environmental flow of the MFM with the available NOAA wind data. There was not any distinct gust data reported to enable a separate analysis of gust speeds on the flow measurements. We have added this figure to the SI that illustrates that the hourly standard deviation of the environmental flow tends to increase with wind speed, and added a reference to the figure in the text of the revised manuscript.

[Figure]

Minor Revisions:

(Lines 17 and 198) - Please check for consistency. Abstract reports at 0C, 1 atm and paper reports 25C, 1 atm.

> KBH: This has been corrected in the revised manuscript. All gas volumes in the paper are referenced to 25 C/1 atm. This mix up is sourced from the varying definitions of STP in the literature, some of which are substantially different than IUPAC. We have elected to be specific about the reference conditions used in the manuscript (and not use acronyms like 'STP' or 'NTP') to eliminate confusion for ourselves and the reader.

(Line 195) "... sequentially decreased through 9 steps to 6.0 x 10^(-5) m^3/h ..." - please list out the steps for increased clarity

> KBH: We have updated the text to list that the flow steps go as approximate factors of 2 over the dynamic range of the MFC.

(Line 231) - Why was the initial condition of the chamber set to 1 ppm CH4 when global background is closer to 2 ppm?

> KBH: 1 ppm methane was chosen as a low concentration end member to avoid divide-by-zero errors during the initial steps of the simulation of the measurement systems rather than to reflect a given deviation from atmospheric background. Practically there is another

issue that the air near point sources is often elevated and further consideration of background/initial concentrations are needed for the dynamic chamber approach.

The text has been updated to explain our rationale.

(Line 232) - System of equations for the simulations need to be listed and added to SI.

KBH: We have updated the SI to discuss how we calculated the evolution of gas concentrations in the MFM and dynamic chamber and how the response functions for each system were simulated.

(Line 311) - "The presence [of] a range..."

KBH: Thank you for finding this typo! It has been corrected.

(Fig. 9) - Can be moved to SI. Does not add much to text results.

KBH: We have moved figure 9 to the SI and renumbered figures in the text.

(Figs. 8 and 9) - Clarity. Needs legend for colors.

KBH: We have improved figures 8 and 9 following this comment and a similar remark from referee number 2 to more clearly explain that the line colors are MFM flow resistance values, and that the arrows reflect how soil water affects permeability, and thus flow through the MFM, and how different aspects of the chamber design and soil interface affect flow. Figure 9 was moved to the SI and the figures are renumbered.

(Line 393) "... simulated environmental flow ..." - Clarity. Was this simulated using Eq. 4?

KBH: Yes, this was simulated with equation 4. A reference to equation 4 has been added.

(Line 430) "We examined a set of environmental extremes ..." - Clarity. Were these extremes plugged into Eq. 4?

KBH: To clarify how the flows were estimated, a reference to equation 4 has been added at this point in the manuscript.

(Table 2) - Reference check Equation 8.

KBH: Thank you for pointing out this oversight, this should be a reference to equation 4.

**RC2**: 'Comment on egusphere-2025-1201', Anonymous Referee #2, 20 Jun 2025

Overall, this study provides some very important and beneficial scientific experimentation on a method of conducting methane emissions measurements from orphaned oil and gas wells (OOGWs). As this research topic has gained traction in the last few years many different methodologies have been developed in rapid succession adapting from previous work with a "this should work" mentality. Which is why studies like this are so beneficial.

> KBH: Thank you for taking the time to review the work and provide input. We are especially motivated by this studies ability to provide clarity and utility for orphan well related science. We have modified the manuscript following your suggestions.

Revisions:

Figure 1: At some point in discussing the rigid chamber the general shape of the chamber should be discussed, as in static chamber measurements this general shape has shown to have an impact on gas mixing within chambers. Figure 1 shows 2 different chamber shapes, the smaller rectangular black chamber and the more cylinder like chambers. Even though it is stated in the supplemental material that the rectangular chamber is used it should be clarified what chamber shape was used for the different tests that took place.

> KBH: To clarify, we have added language specifying that the chamber used in this experiment was cuboid in shape at the beginning of Section 4 (Methods). We also added a sentence in the conclusions that considering mixing effects is important to ensure the gas leaving the chamber matches the target gas the MFM was calibrated for.

Figure 8: The compass arrows on the graph are confusing, I think it might be cleaner if the horizontal and vertical arrows aren't intersecting with each other.

> KBH: We have revised figure 8 and figure 9 in the manuscript to have a legend defining the values of each of the lines and have separated and labeled the separate arrows to more clearly explain that they are intended to illustrate how the gas flows through the soil and MFM changes with soil properties and chamber design criteria. As noted above we have moved figure 9 in the original manuscript into the SI and renumbered the figures. The informational value of this figure has been considerably improved.

Line 371: It is mentioned that the chamber was deployed during rain events, I'm guessing that these rain events coincide with the changes in chamber water vapor pressure. If data is readily available a supplemental table showing amount of precipitation during each day of the testing would help clarify this assumption.

> KBH: Near line 384 we have added a sentence to the manuscript stating precipitation occurred on two dates, 2024/01/06 and 2024/01/09 as day-long rain events. A change in water vapor pressure can be observed on 2024/01/06 during both events, but they are also coupled with temperature driven changes in water vapor pressure.

Line 254 & Line 510: Beginning line 254, "Ultimately, it was determined that a chamber placement depth of 18 cm (7 inches) or more, removing fibrous and porous debris from the soil before packing (roots, sticks, rocks, and pebbles, etc.) and wetting the soil packed at the edges, provided a reliable seal that could be blind tested against reference flows and could achieve reliable results." Beginning Line 510, "The edges of the rigid chamber need to be sealed against the ground through a combination of deep burial (7 cm or more in our findings), soaking the soil with water to encourage settling and reducing permeability, and adding weights on top of the chamber or other measures to prevent the chamber from shifting and breaking the soil seal." Where did 7cm come from as a minimum for chamber edge burial if it was determined that depth of 18 cm was required for reliable blind tests? Does the suggestion of 7cm have more to do with the wide variability of soil types present in the potential deployment scenarios of chambers?

> KBH: First, we apologize for this confusing set of statements. A 7 cm depth finding was not our intention. The need to convert inches (the units of the tape measure in the field) to cm resulted in a mix up, and this has been addressed: we found, for our soil plot, that 18 cm (7 inches) (*not 7 cm*) was required for 3 sequential successful placements. The manuscript has been corrected at line 515 to have the true units. Thank you for catching this mistake.
>
> The referee raises an excellent point: We are not trying to prescribe a specific depth to users. Our goal is to illustrate and emphasize that use of a rigid chamber and mass flow meter is extremely sensitive to the quality of the ground seal due to the requirement that the chamber contain the pressure required to drive the leak through the MFM, as well as the need to create a sufficiently high flow resistant in the soil to cause flow to be preferential through the MFM. In our testing and as shown in figure 6, any leak can result in substantial loss of signal that may not be apparent to an unwary user (in our testing this failure mode was almost binary in nature, either full recovery of the reference flow was achieved, or the signal was almost undetectable due to a leak). The depth we found for soil plot is notably deeper and laborious to achieve than some static or dynamic chambers we have seen, and that the ground seal is easily disrupted by chamber motion, however, small. We have rephrased the start of our recommendation (~line 515 in the original manuscript) to highlight the priority is a consistent leak-free seal, and the 18 cm depth in our plot is illustrative, and it does not seem like shallower depths will generate reliable results.

KBH: In addition to the changes suggested by the referees, we have also made several changes to the revised manuscript:

(1) We have changed the title to read "Evaluating mass flow meter measurements from chambers for greenhouse gas emissions from orphan wells and other point sources".
(2) We have modified the colors in plots 6, 7, and 9 to match EGU accessibility guidelines.